# Experimental Evidence Reveals Both Cross-Infection and Cross-Contamination Risk of Embryo Storage in Liquid Nitrogen Biobanks

**DOI:** 10.3390/ani10040598

**Published:** 2020-04-01

**Authors:** Clara Marin, Ximo Garcia-Dominguez, Laura Montoro-Dasi, Laura Lorenzo-Rebenaque, José S. Vicente, Francisco Marco-Jimenez

**Affiliations:** 1Instituto de Ciencias Biomédicas, Facultad de Veterinaria, Universidad Cardenal Herrera-CEU, CEU Universities, Avenida Seminario s/n, 46113 Moncada, Spain; clara.marin@uchceu.es (C.M.); laura.lorenzo@alumnos.uchceu.es (L.L.-R.); 2Instituto de Ciencia y Tecnología Animal, Universitat Politècnica de València, 46022 Valencia, Spain; ximo.garciadominguez@gmail.com (X.G.-D.); laura.montoro@outlook.com (L.M.-D.); jvicent@dca.upv.es (J.S.V.)

**Keywords:** microorganisms, pathogen transmission, vitrification, bacteria, fungi, embryo, naked device, closed device

## Abstract

**Simple Summary:**

This study was conducted to demonstrate the potential hazards of cross-infection and cross-contamination of embryos during storage in liquid nitrogen biobanks. For the harmless and successful cryopreservation of embryos, the vitrification method must be chosen meticulously to guarantee not only a high post-thaw survival of embryos, but also to reduce the risk of disease transmission when those embryos are in storage for long periods.

**Abstract:**

In recent decades, gamete and embryo cryopreservation have become routine procedures in livestock and human assisted reproduction. However, the safe storage of germplasm and the prevention of disease transmission continue to be potential hazards of disease transmission through embryo transfer. This study aimed to demonstrate the potential risk of cross-infection of embryos from contaminated liquid nitrogen, and cross-contamination of sterile liquid nitrogen from infected embryos in naked and closed devices. Additionally, we examined the effects of antibiotic-free media on culture development of infected embryos. The study was a laboratory-based analysis using rabbit as a model. Two experiments were performed to evaluate both cross-infection (liquid nitrogen to embryos) and cross-contamination (embryos to liquid nitrogen) of artificially inoculated *Salmonella* Typhimurium, *Staphylococcus aureus*, *Enterobacter aerogenes*, and *Aspergillus brasiliensis*. Rapid cooling through vitrification was conducted on rabbit embryos, stored for a year, thawed, and cultured. In vivo produced late morulae–early blastocyst stages (72 h) embryos were used (n = 480). Embryos were cultured for 1 h in solutions with and without pathogens. Then, the embryos were vitrified and stored in naked and closed devices for one year in two liquid nitrogen biobanks (one pathogen-free and the other artificially contaminated). Embryos were warmed and cultured for a further 48 h, assessing the development and the presence of microorganism (chromogenic media, scanning electron microscopy). Embryos stored in naked devices in artificially contaminated liquid nitrogen became infected (12.5%), while none of the embryos stored in closed devices were infected. Meanwhile, storage of artificially infected embryos incurred liquid nitrogen biobank contamination (100%). Observations by scanning electron microscopy revealed that all the microorganisms were caught in the surface of embryos after the vitrification-thawed procedure. Nevertheless, embryos cultured in antibiotics and antimycotic medium developed to the hatched blastocyst stage, while artificially infected embryos cultured in antibiotic-free medium failed to develop. In conclusion, our findings support that both cross-contamination and cross-infection during embryo storage in liquid nitrogen biobanks are plausible. So, to ensure biosafety for the cryogenic storage, closed systems that avoid direct contact with liquid nitrogen must be used. Moreover, it seems essential to provide best practice guidelines for the cryogenic preservation and storage of gametes and embryos, to define appropriate quality and risk management procedures.

## 1. Introduction

Nowadays, gamete and embryo cryopreservation are routine procedures in livestock and human assisted reproduction [1,2,3]. Nevertheless, the safe storage of germplasm and prevention of disease transmission is a concern for animal and human health [4]. Disease transmission has been described during cryopreservation procedures [5,6]. The origin of the microbial infection can be both systemic and local infections of the reproductive tract, as well as the inadvertent introduction of microorganisms during germplasm processing [7]. Several critical factors may influence the contamination of embryos with pathogens during cryopreservation, including the integrity of the embryonic pellucid zone, the cooling method, loading, and sealing of the freezing container, and the sterility of the liquid nitrogen and the Dewar storage container. In this sense, pathogen survival during embryo cryopreservation and storage at cryogenic temperatures has been proven [4,6,8,9]. In the absence of securely sealed closed-device systems, there is a theoretical risk of microbial and viral cross-contamination between specimens commonly stored in liquid nitrogen [5,10,11,12]. Liquid nitrogen presents significant risks in the transmission and propagation of diseases [13,14]. Components in culture and cryoprotective additives may act as a stabilizer or prevent cryoinjuries for microorganisms at cryogenic temperatures [5,14,15]. So, the risk of transfer of microorganisms from liquid nitrogen to stored gametes and embryos is notable. Previous studies demonstrated the possibility of cross-infection from infected embryos to sterile ones within liquid nitrogen [16]. Different bacterial species have been isolated during storage in liquid nitrogen, such as *Stenotrophomonas maltophilia*, *Bacillus* spp., *Enterobacter* spp., and *Staphylococcus* spp., some of them related to the suppression of embryonic development [4,6]. Thus, to minimise the risks of microbial growth, antibiotics have been introduced at sample collection, fertilisation, culture, or storage [17,18,19]. However, the opposite effect of antibiotics on the growth rate of preimplantation embryos has been well documented [19,20,21]. Taking all these data into account, the bacterial cross-infection risk in embryo cryopreservation is deemed a significant public health concern [22]. 

The aim of this study was to evaluate the risk of cross-infection of sterile embryos from contaminated liquid nitrogen, and cross-contamination of sterile liquid nitrogen from infected embryos, using naked and closed devices stored for one year. Additionally, we examined the effects of antibiotic-free media on culture development of infected embryos.

## 2. Materials and Methods 

All the experimental procedures used in this study were performed following Directive 2010/63/EU EEC for animal experiments and were reviewed and approved by the Ethical Committee for Experimentation with Animals of the Universitat Politècnica de Valéncia, Spain (research code: 2015/VSC/PEA/00061).

### 2.1. Experimental Design

Two experiments were performed simultaneously to evaluate both cross-infection capacity (liquid nitrogen to embryos) and cross-contamination capacity (embryos to liquid nitrogen) of pathogenic microorganisms (*Salmonella* spp., *Staphylococcus* spp., *Enterobacter* spp., and *Aspergillus* spp. fungus) in two biobanks (11 L, liquid nitrogen aluminium Dewar MVE SC11/7). Embryos were cryopreserved using different vitrification devices: Cryotop device (Kitazato Co., Fuji, Japan) closed and naked, and French mini-straw (IMV, France). A schematic of the steps of the experiments carried out is provided in Figure 1. Before performing both experiments, liquid nitrogen biobanks were washed and disinfected with a 5% chlorine solution and tested to confirm the absence of pathogens used in this study by culture method (see section on microbiological procedures below). In each session, all media culture and fluids involved in the process of collecting and storing the embryos were tested for sources of contamination, and any positives were discarded.

### 2.2. Experiment 1. Cross-Infection: From Artificially Contaminated Liquid Nitrogen to Embryos

In this experiment, cross-infection between artificial contaminated liquid nitrogen and sterile embryos was evaluated. For this purpose, four commercial strains (three bacteria and one fungus) were used. Specifically, the strains used were *Salmonella* Typhimurium (*S.* Typhimurium; ATCC® 14028™/WDCM 00031, Scharlab, S.L.); *Staphylococcus aureus* (*S. aureus*; ATCC® 6538™/WDCM 00032, Scharlab, S.L.); *Enterobacter aerogenes* (*E. aerogenes*; ATCC® 13048™/WDCM 00175, Scharlab, S.L.), and *Aspergillus brasiliensis* (*A. brasiliensis*; ATCC® 16404™/WDCM 00053, Scharlab, S.L.) provided by Spanish Type Culture Collection (CECT). Growth and propagation of bacteria and fungus strains were performed according to manufacturer recommendation at an infective titre of 10^6^ CFU/mL for each microorganism. Then, 1 mL of each microorganism was atomised directly into the liquid nitrogen in the half-full biobank. After that, the biobank was filled and stored for one week until the embryos were incorporated. The infective titre was in the order of 10^2^ CFU/mL.

### 2.3. Embryo Recovery and Vitrification

Seven nulliparous New Zealand White does were superovulated with a combination of FSH (Corifollitropin alpha, 3 µg, Elonva, Merck Sharp & Dohme, S.A.) and hCG (7.5 UI). Sixty hours after superovulation, does were inseminated with pooled semen from New Zealand bucks of proven fertility. Ovulation was induced with 1 µg buserelin acetate (Suprefact; Hoechst Marion Roussel, S.A., Madrid, Spain). Females were euthanised 72 h after artificial insemination, and the reproductive tract was immediately removed. Embryos were recovered by flushing each uterine horn with 10 mL Dulbecco´s phosphate-buffered saline containing 0.2% (wt/vol) bovine serum albumin (BSA) at 25 °C. Collected embryos were counted and evaluated following the International Embryo Technology Society (IETS) criteria [23]. Briefly, only embryos in late morulae–early blastocyst stages with homogenous cellular mass and spherical mucin coat and zona pellucida were catalogued as suitable (transferable) embryos. A total of 236 embryos were pooled and vitrified according to the methodology described by Marco-Jiménez et al. [24], using Cryotop and French mini-straw. A group of five to six embryos were located on each vitrification device. The embryos were vitrified in a two-step addition procedure. At vitrification time, embryos were transferred into equilibration solution consisting of 10% (vol/vol) ethylene glycol and 10% (vol/vol) dimethyl sulfoxide dissolved in base medium (BM; Dulbecco’s phosphate-buffered saline containing calcium chloride [9.0 mM] and magnesium chloride [4.9 mM], and supplemented with 0.2% [wt/vol] BSA at pH 7.2) at room temperature (22–25 °C) for 2 minutes. The embryos were then transferred to vitrification solution consisting of 20% (vol/vol) ethylene glycol and 20% (vol/vol) dimethyl sulfoxide in BM, loaded into the devices, and directly plunged into liquid nitrogen within 1 minute. Cryotop devices were vitrified in direct contact with pathogen-free liquid nitrogen. Then, Cryotop devices were divided into two experimental groups: Cryotop within a protective cap that isolates the loaded sample from the cryogenic fluid (closed system), and Cryotop maintained without protective cap during storage (naked system). Liquid nitrogen was provided by Air Liquide, coming from a sterile cryogenic distillation process from the previously filtered air. Routine maintenance included the addition of new liquid nitrogen once a week to maintain the level. Samples were stored for a year.

### 2.4. Experiment 2. Cross-Contamination: From Artificially Infected Embryos to Liquid Nitrogen

In this experiment, cross-contamination between artificially infected embryos and liquid nitrogen and cross-infection between embryos were evaluated. For this purpose, a total of 244 rabbit embryos from seven females were obtained as previously described and used in this experiment. Embryos were pooled (five to six embryos per pool) and each pool sample was inoculated with one microorganism (*S.* Typhimurium, *S. aureus*, *E. aerogenes* and *A. brasiliensis*) at an infective titre of 10^6^ CFU/mL (prepared as reported above). Then, each pool was placed in a 4-well plate Nunc (500 µL of medium) and incubated in BM solution with the microorganism for 1 h at room temperature (22–25 °C). Non-infected embryos were also included in a control group. After that, embryos were washed in two consecutive steps in BM. Then, embryos for each experimental group (infected and control) were distributed in two groups to be vitrified in Cryotop-naked (n = 3) and Cryotop-closed with straw cap (n = 3) devices. Then, a group of five to six embryos were located on each vitrification device. In each canister in the biobank, only devices containing embryos infected by the same microorganism or control embryos were stored. All devices were stored as previously. Likewise, routine maintenance included the addition of new liquid nitrogen once a week to maintain the level. Samples were stored for a year.

### 2.5. Detection of Infectious Agents

#### 2.5.1. Sample Collection (Experiment 1 and Experiment 2)

After 1 year of storage, both liquid nitrogen and embryos were evaluated. The collection of liquid nitrogen samples was carried out following guidelines described by several authors [4,6], which enabled us to obtain liquid nitrogen from the bottom of the containers. For that purpose, liquid nitrogen was withdrawn aseptically from each container using a 50 mL sterile centrifuge tube (tube with a hole in the cap) and evaporated at room temperature in a biological safety cabinet. To resuspend dried material present in the centrifuge tube, 5 mL of thioglycolate broth was added to each sample with thorough shaking. A total of nine samples were taken (four and five for experiments 1 and 2, respectively). Throughout the year of storage, 11 samples of liquid nitrogen were collected from the supplier to rule them out as a source of contamination after refilling.

After storage in liquid nitrogen, embryos stored in Cryotop were warmed by sudden immersion of the naked devices in 200 µL drops of 0.33 M sucrose at 25 °C in BM. After 5 minutes, the embryos were washed in BM. Embryos contained in mini-straw were thawed in two steps, placing the mini-straw 10 cm horizontally from liquid nitrogen vapour for 20–30 s, and when the crystallisation process began, the mini-straw was immersed in a water bath at 25 °C for 10–15 s. Then, the entire content was tipped out into a plate containing 0.33 M sucrose solution in BM for 5 minutes. Then, embryos were cultured in 500 µL of Tissue Culture Medium 199 without antibiotics and antimycotic at 38.5 °C, 5% CO_2_, and saturated humidity for 1 h. After that, embryos were moved to a medium containing 1% penicillin-streptomycin and amphotericin B, and cultured for 47 h to assess their development ability until hatching/hatched blastocyst stage (proportion of hatching and hatched blastocyst at 48 h of culture from total cultured embryos). The hatched state was achieved when more than 50% of the embryonic cell mass was extruded from the zona pellucida. The medium without antibiotics was used to evaluate viable bacterial and fungal burden. Some embryos were cultured as previously, but in antibiotic- and antimycotic-free medium. 

#### 2.5.2. Microbiological Procedures

All samples collected during the experiments were tested for the presence of applied microorganisms, and analyses were carried out under a laminar flow hood using standard microbiological methods. To evaluate the presence and viability of the different microorganisms in the samples (viable and culturable), 100 µL of suspension from each sample was inoculated onto selective media for each microorganism: XLT-4 Agar (Xylose Lactose Tergitol 4, Scharlau, Barcelona, Spain) for *S.* Typhimurium detection; Baird Parker Agar (Scharlau, Barcelona, Spain) for *S. aureus* detection; Cetrimide Agar (Scharlau, Barcelona, Spain) for *E. aerogenes* detection; and Sabouraud Chloramphenicol Agar (Scharlau, Barcelona, Spain) for *A. brasiliensis* detection. Culture plates were incubated for 24–48 hours at 37 °C. After incubation, suspected bacteria colonies were streaked into a nutrient medium (Scharlab, S.L., Barcelona, Spain) and incubated at 37 °C for 24 hours. Then, API-test (Biomerieux, S.L., Barcelona, Spain) was performed to confirm the bacteria obtained. Filamentous fungi were identified on the basis of macroscopic and microscopic morphologic features (Figure 2).

### 2.6. Scanning Electron Microscopy (SEM)

To demonstrate bacterial and fungus attachment after the vitrification-thawed procedure, the mucin coat of embryos was examined by SEM. To this end, one sample of six embryos infected by each microorganism was visualised. In brief, embryos were fixed in 2.5% glutaraldehyde in DPBS at 4 °C. They were washed three times in DPBS, rinsed in deionised water and then post-fixed in 1% osmium tetroxide in deionised water for 1.5 h. After rinsing in deionised water, embryos were dehydrated in an ethanol series, critical point dried (Leica CPD300), sputter-coated with platinum on a Leica EM Med 020 and examined on a Zeiss ULTRA 55 scanning microscope.

### 2.7. Statistical Analysis

A generalised linear model was used, including the vitrification devices (Cryotop-naked, Cryotop-closed and mini-straw devices) and microbiological contamination as fixed effects. The error was designated as having a binomial distribution using a probit link function. Binomial data for hatching/hatched blastocyst rate were assigned a 1 if positive development had been achieved, or a 0 if it had not. A P value of less than 0.05 was considered to indicate a statistically significant difference. The data are presented as least square means ± standard error mean. All statistical analyses were carried out using a commercially available software program (SPSS 21.0 software package; SPSS Inc., Chicago, IL, USA, 2002).

## 3. Results

All media culture and fluids involved in the process of collecting and storing the embryos before the artificial contamination process tested negative for microorganisms analysed in this study. In addition, collected liquid nitrogen samples also tested negative.

### 3.1. Experiment 1. Cross-Infection: From Contaminated Liquid Nitrogen to Embryos

A total of six liquid nitrogen samples were analysed. All the samples were found to be positive (viable-culturable) for *S. aureus*, *E. aerogenes*, and *A. brasiliensis* (Figure 2). However, none of the samples were positive for *S*. Typhimurium. None of the embryos stored in closed devices (Cryotop and mini-straw) were infected. However, six embryos stored in one Cryotop-naked (12.5%) were infected with *S. aureus*. A total of 236 embryos were analysed (211 cultured in antibiotics and antimycotic medium, and 25 cultured in antibiotic- and antimycotic-free medium). As shown in Table 1, rates of embryo development to the hatched blastocyst stage showed no significant differences between all vitrified groups.

### 3.2. Experiment 2. Cross-Contamination: From Infected Embryos to Liquid Nitrogen

A total of 30 embryos were processed for SEM evaluation. Observations by SEM after vitrifying-thawing and washing procedure to be cultured revealed that all the microorganisms were caught in the surface of the embryos (Figure 3). 

A total of three liquid nitrogen samples were analysed. All the samples were found to be positive (viable-culturable) for *S. aureus* (100%), and two for *E. aerogenes* (66.6%). None of the liquid nitrogen samples were positive for *S.* Typhimurium and *A. brasiliensis* after a year of storage. After embryo thawing, all microorganisms (*S.* Typhimurium, *S. aureus*, *E. aerogenes*, and *A. brasiliensis*) survived (viable-culturable). No cross-infection was isolated in embryo pools between Cryotop-naked and Cryotop-closed. Likewise, no infection was detected in any control sample of Cryotop-naked and Cryotop-closed. A total of 244 embryos were analysed (224 cultured in antibiotics and antimycotic medium, and 20 cultured in antibiotic- and antimycotic-free medium). As shown in Table 2, rates of embryo development to the hatched blastocyst stage showed no significant differences between Cryotop-naked and Cryotop-closed. Culture dishes were colonised by microorganism in antibiotic-free medium, affecting embryonic development (Figure 2). 

## 4. Discussion

We set out to evaluate our hypothesis that cross-contamination and cross-infection are plausible risks associated with embryo storage in liquid nitrogen biobanks, although the peril depends on the pathogen. Even though embryo storage in liquid nitrogen can be a potentially hazardous situation where biological material is kept together [13], the vast majority of studies have theorised about that [10,25]. However, no studies address this issue by using experimental conditions to avoid bias.

The risk of infection of germplasm stored in a biobank has received a high degree of attention [7,13]. In this sense, the risk of contamination of liquid nitrogen by common environmental pathogens has been widely described [4,6]. In line with this, our data confirms the occurrence of both cross-contamination and cross-infection again after long-term storage of embryos in liquid nitrogen biobanks using naked system devices. In addition, our data indicates that species of environmental pathogens such as *S. aureus* and *E. aerogenes* survive attached to cryopreserved embryos, but also when they come into direct contact with liquid nitrogen. Several authors describe how, under assisted reproductive technology laboratory conditions, pathogens can survive at low temperatures and may contaminate the biologic material or surface of the biobanks [13]. Hence, the use of naked system devices for germplasm storage compromises the sterility, due to the direct exposure to liquid nitrogen during vitrification and storage [6,13]. In a previous study, where liquid nitrogen was experimentally contaminated with *Escherichia coli* and *Staphylococcus aureus*, 94% of semen pellets showed positive infection [26]. This is explained by the double role of the liquid nitrogen biobanks during routine replenishment. Firstly, replenished liquid nitrogen biobanks presented increased microbial concentration [16]. Secondly, replenished liquid nitrogen biobanks generate liquid nitrogen movement, thus facilitating the dispersion and contact of microorganisms with stored biological material [8]. And if to this we add that the most crucial concept of the gametes vitrification method, i.e., that the reduction in the volume of cryoprotectant, allows for a greater rate of cooling by direct contact with the liquid nitrogen [10,27], the risk seems obvious. However, minimum essential volume devices offer high survival and developmental rates in the vast majority of species [24]. In this study, in vitro embryo culture development was similar between devices. Several authors have reported the same results in embryo survival rates and pregnancy outcomes between open and closed devices [10,28,29,30]. In oocyte vitrification, a decrease in cryosurvival rate due to the use of closed devices has been reported [31]. However, Cryotop and mini-straws had a similar embryonic development in rabbit species [24], but slightly less than the in vitro development, compared with fresh embryos [24,32,33]. Bielanski et al. [13] showed that Cryotop-closed with a straw cap system and straws had the same embryo development results after a year of storage, with a lower potential danger of embryo contamination. In a recent study comparing open versus closed with straw cap vitrification systems, no differences in the risk of cross-contamination between devices were described [6]. In part, this would be due to the difference in infectivity level or the microorganisms evaluated, as well as different storage conditions compared to the present study. 

The use of aseptic closed vitrification devices with straw cap (Cryotop and straw) or liquid nitrogen sterilisation provides a reasonable alternative to avoid the contamination [34,35]. Nevertheless, closed vitrification devices do not prevent persistence of the embryo infection. Components in culture medium and cryoprotective additives may act as a stabiliser for microorganisms at cryogenic temperatures [13,14,15]. Many researchers have recognised that embryo collection procedures and cryopreservation protocols are not sterile techniques [4,6]. Hence, our results confirm that if embryos are infected after vitrification and warming procedures, they maintain their potential infection capacity. Cross-contamination of cryopreserved tissue was first reported by Tedder [36]. Nowadays, microbiological transmission and embryo infection, especially with environmental pathogens that retain their viability for long periods at cryogenic temperatures, are a growing public health concern [13]. The demonstrated infection of embryos and contamination of liquid nitrogen by *E. aerogenes* and *S. aureus* is particularly relevant for reproductive contexts, given the implications this may entail in embryo transfer approaches and embryo health certification for international movement [13]. In fact, there is a multitude of factors and conditions that affect microorganism surveillance [15]. It is known that common microorganisms are contaminating cryobanks. As an example, *Salmonella* spp., the second most common zoonotic disease, can survive during gamete cryostorage procedures [37]. *A. brasiliensis*, an environmental pollutant, can survive in the debris of liquid nitrogen storage containers [4]. Moreover, *E*. *aerogenes*, the most common cause of environmental contamination due to their survival capacity in extreme environments, are able to survive in direct contact with liquid nitrogen [38]. *S. aureus* can provoke cross-contamination between liquid nitrogen and embryos stored in the Cryotop-naked device, and the risk is higher because it is one of the most important zoonotic pathogens, due to its high versatility and adaptability [39,40]. Furthermore, *Staphylococus* spp. can colonise the reproductive systems of, and cause abortions in, cattle [41,42] and cause human infertility [43]. Based on our results, *S. aureus* is especially relevant due to its capacity for survival and spread in frozen storage [44]. The cold shock resistance of this pathogen is well documented [45]. The translocation capacity of *S. aureus* from contaminated semen in sterile liquid nitrogen after two hours of exposure has also been described [26]. Additionally, Bielanski et al. [4] showed the contamination of *Staphylococcus* spp. in liquid nitrogen and embryo after 15 years of storage. This ability may be explained by their capacity for the production of biofilms that protect the bacteria [46]. As proven by the present work, the survival capacity and cross-contamination between liquid nitrogen and stored embryos in the naked system device depict a high risk due to the bacteria’s capacity to survive [47]. To prevent and control bacterial infection, antibiotics are routinely added to the embryo culture media, with penicillin, streptomycin, and gentamycin being commonly used [21,48]. To evaluate the antibiotic role, we included them in the medium during embryo culture. As proposed by Bielanski [7], our SEM imaging shows that in embryos that were infected, the microorganism remained firmly adhered to the surface after the vitrification procedure, which could have consequences in terms of microorganism spreading.

Therefore, antibiotics are required for further embryo development, as they prevent bacterial growth and culture media contamination. With time, widespread bacterial antimicrobial resistance diminishes the clinical efficacy of antibiotics, threating the health of humans and animals [49,50]. There is serious concern about the rise of antibiotic-resistant “superbugs”, some of which are now impervious to many antibiotics [51]. These include the ESKAPE pathogens (*Enterococcus faecium, Staphylococcus aureus, Klebsiella pneumoniae, Acinetobacter baumannii, Pseudomonas aeruginosa*, and *Enterobacter* spp. [52]. Henceforth, *Pseudomonas aeruginosa* (defined as critical) and *Staphylococcus aureus* (defined as high) have been included in the World Health Organisation priority list of 12 antibiotic-resistant bacteria [38,53]. Besides this, livestock-associated methicillin-resistant *Staphylococcus aureus* remains an emerging problem around the world [54]. Over 90% of the bacteria isolated from seminal doses are resistant to common antibiotics added to extenders [55]. If this trend continues, the consequences for public health and the general community could be catastrophic [56]. To overcome this problem, the identification of therapeutic agents that can provide alternative treatments against conventional antibiotics are required.

## 5. Conclusions

Our findings clearly support that both cross-contamination and cross-infection during embryo storage in liquid nitrogen biobanks are plausible. So, to ensure biosafety for cryogenic storage, the use of closed systems that avoid direct contact with liquid nitrogen must be considered. Moreover, it seems essential to provide best practice guidelines for the cryogenic preservation and storage of gametes and embryos, in order to define appropriate quality and risk management procedures.

## Figures and Tables

**Figure 1 animals-10-00598-f001:**
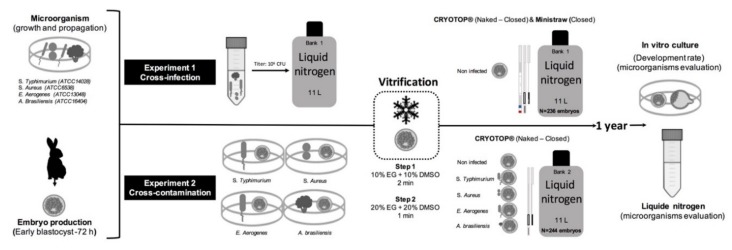
Schematic diagram of the experiments carried out to evaluate both cross-contamination and cross-infection risk of embryo storage in liquid nitrogen biobanks. In experiment 1, cross-infection between artificially contaminated liquid nitrogen with 10^6^ CFU of *Salmonella* Typhimurium, *Staphylococcus aureus*, *Enterobacter aerogenes* and *Aspergillus brasiliensis* and vitrified embryos stored in Cryotop (closed and naked) and mini-straw devices were assessed. In experiment 2, cross-contamination between artificially infected fresh embryos, vitrified and stored in Cryotop (closed and naked) and liquid nitrogen was assessed.

**Figure 2 animals-10-00598-f002:**
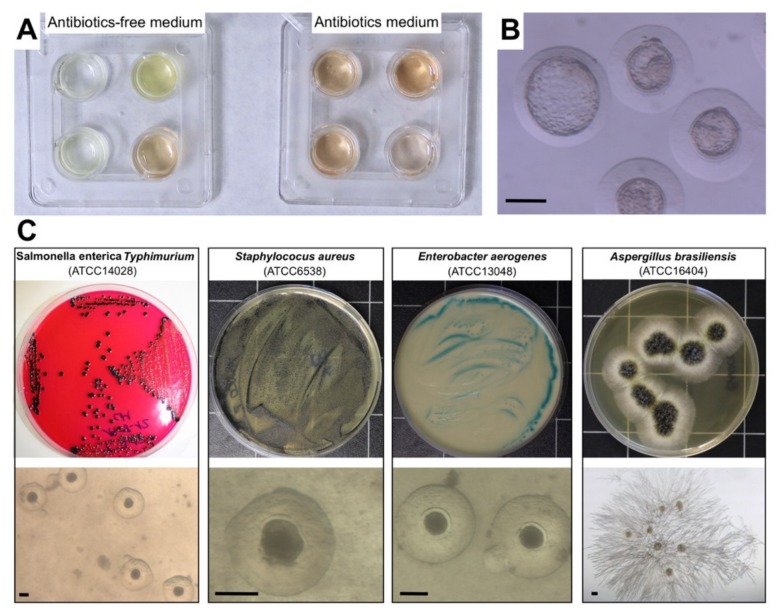
Representative images of microorganism viability. (**A**) Plates containing medium after embryo culture for 48 h in antibiotic- and antimycotic-free medium (left) and in antibiotic and antimycotic medium (right). (**B**) Non-infected vitrified-thawed embryos cultured for 48 hours in antibiotic- and antimycotic-free medium. (**C**) Representative sample image of positive plates after inoculated 100 µL of culture medium on selective media and vitrified-thawed embryos cultured for 48 hours in antibiotic- and antimycotic-free medium. *S.* Typhimurium sample cultured onto XL. *S. aureus* sample cultured onto Baird Parker Agar. T-4 Agar. *E. aerogenes* sample cultured onto Cetrimide Agar. *A. brasiliensis* sample cultured onto Sabouraud Chloramphenicol Agar. Scale bar: 150 µm.

**Figure 3 animals-10-00598-f003:**
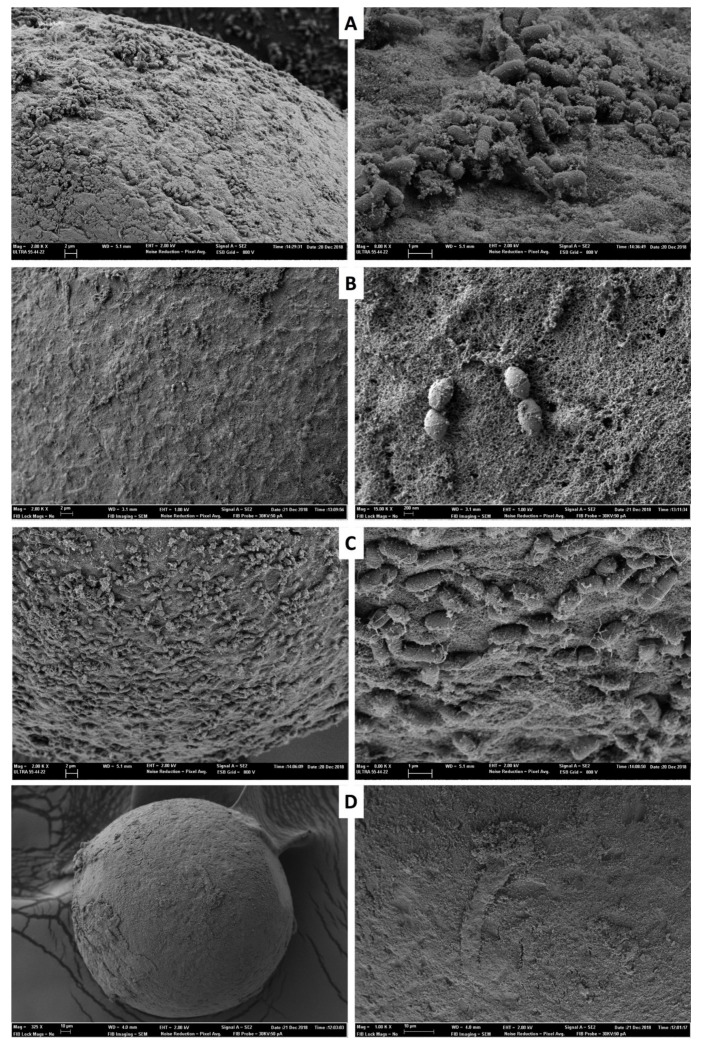
Representative scanning electron microscopy of rabbit embryo surfaces infected with pathogens after vitrification-thawed procedure. (**A**) *Salmonella enterica* Typhimurium (ATCC14028) (magnification of 2.00 KX, left and 8.00 KX, right). (**B**) *Staphylococus aureus* (ATCC6538) (magnification of 2.00 KX, left and 15.00 KX, right. (**C**) *Enterobacter aerogenes* (ATCC13048) (magnification of 2.00 KX, left and 8.00 KX, right. (**D**) *Aspergillus brasiliensis* (ATCC16404) (magnification of 325 X, left and 1.00 KX, right).

**Table 1 animals-10-00598-t001:** In vitro development of rabbit embryos at the morula stage after 1 year of storage in liquid nitrogen, experimentally infected with *Salmonella* Typhimurium, *Staphylococcus aureus*, *Enterobacter aerogenes,* and *Aspergillus brasiliensis* biobank using three cryostorage carriers.

Device	n	Hatching/Hatched Blastocyst Rate
Cryotop-naked	57	71.0 ± 5.10
Cryotop-closed	67	76.0 ± 4.50
Mini-straw	87	82.0 ± 3.70

n: Number of embryos. Data are presented as least squares means ± standard error of the least squares means.

**Table 2 animals-10-00598-t002:** In vitro embryo development acquired after experimental infection of rabbit embryos at the morula stage after 1 year of storage in a microorganism-free liquid nitrogen biobank, using two cryostorage carriers.

Device	Pathogen	n	Hatching/Hatched Blastocyst Rate
Cryotop-naked	*Salmonella* Typhimurium	23	78.0 ± 8.6
	*Staphylococcus aureus*	24	71.0 ± 9.3
	*Enterobacter aerogenes*	22	77.0 ± 8.9
	*Aspergillus brasiliensis*	24	79.0 ± 8.3
	Control	14	86.0 ± 9.4
Cryotop-closed	*Salmonella* Typhimurium	19	73.0 ± 11.1
	*Staphylococcus aureus*	24	79.0 ± 8.3
	*Enterobacter aerogenes*	25	82.0 ± 5.4
	*Aspergillus brasiliensis*	25	74.0 ± 9.6
	Control	24	79.0 ± 8.3

n: Number of embryos. Data are presented as least squares means ± standard error of the least squares means.

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
