# Peer review of "Experimental Evidence Reveals Both Cross-Infection and Cross-Contamination Risk of Embryo Storage in Liquid Nitrogen Biobanks"

_animals, 2020, doi:10.3390/ani10040598_

Round 1

Reviewer 1 Report

The authors are providing clear evidence of the risks related to microorganisms during embryo storage in liquid nitrogen. I consider it a sound scientific work. There are minor suggestions that I would like to address:

Abstract: I suggest to insert the microorganisms that were used in the study.

Keywords: as "cross-infection" and "cross-contamination" are in the title, the authors should select different keywords to replace both.

L135: It is important to clarify why the authors selected 106 UFC/ML for all microorganisms used. 

L172: 100L?

L185: B to E - the explanation is confusing. Please, review it.

          J - Please insert "non-infected"

Author Response

Thank you for your minor suggestions and kind words.

Abstract: I suggest to insert the microorganisms that were used in the study.

All microorganisms has been included in the abstract.

Keywords: as "cross-infection" and "cross-contamination" are in the title, the authors should select different keywords to replace both.

Both keywords has been replaced by Microorganisms and Pathogens transmission. Besides, Liquid Nitrogen has been replace by Vitrification

L135: It is important to clarify why the authors selected 106 UFC/mL for all microorganisms used. 

In this experiment, we used an infective titer of 106CFU/mL based on the environmental contamination described in literature (Bennemann et al., 2018; Navab-Daneshmand et al., 2018; Bajerski et al., 2020). We add 1 mL of each microorganism to 11 L of liquid nitrogen, then infected titer was on the order of 102CFU/mL (at a concentration level close to the detection limit).In our opinion an infectious titer quite realistic. The infective titer was included in the manuscript.

L172: 100L?

It was a mistake. A total of 100 µL of suspension from each sample was inoculated

L185: B to E - the explanation is confusing. Please, review it.

Figure legend has been clarified

Reviewer 2 Report

I think that indication of LN2 without abbreviation it could be better. Also, LN2 is used in the text but it becomes NL2 in figure 1

Line 87: This sentence is not clear “Cryotop device (Kitazato Co., Fuji, Japan) 87 with a straw cap or maintain naked during storage and French mini-straw were judged.”

Figure 1 describes the whole experiment as figure but legend of the figure should be explained well in order to follow the schematic diagram that they are showing. Maybe indicating the type of device that it was used for cryopreservation would improve the understanding of the scheme

Line 120: can you be more clear on how the base medium was done?

Line 113: please add a reference here on the IETS criteria

Line 124: how many embryos did you allocate per cryotop or mini-straw? Indicate please

Line 172: 100 L of suspension = 100 Liters???

Figure 2J: it would be nice if you can show the hatching/fully hatched embryos instead since you got a good rate of embryo hatching

In both experiment 1 and 2, the number of technical replicates should be reported. How many times did you replicate the experiment? Did you do it in different days?

In the discuss it could be important to compare and report the results of embryo hatching of the thawed embryos with data on hatching of fresh embryos from literature. This is something that would add value to the manuscript

Author Response

Comments and Suggestions for Authors

I think that indication of LN2 without abbreviation it could be better. Also, LN2 is used in the text but it becomes NL2 in figure

LN2 acronym has been replaced by liquid nitrogen across the manuscript. Also, NL2 has been replaced by liquid nitrogen.

Line 87: This sentence is not clear “Cryotop device (Kitazato Co., Fuji, Japan) 87 with a straw cap or maintain naked during storage and French mini-straw were judged.

The phrase has been re-written. Embryos were cryopreserved using different vitrification devices: Cryotop device (Kitazato Co., Fuji, Japan) closed and naked and French mini-straw (IMV, France).

Figure 1 describes the whole experiment as figure but legend of the figure should be explained well in order to follow the schematic diagram that they are showing. Maybe indicating the type of device that it was used for cryopreservation would improve the understanding of the scheme

The figure legend has been clarified as: “Figure 1. Schematic diagram of the experiments carried out to evaluate both cross-contamination and cross-infection risk of embryos storage in liquid nitrogen biobanks. In experiment 1, cross-infection between artificial contaminated liquid nitrogen with 106CFU of Salmonella Typhimurium, Staphylococcus aureus, Enterobacter aerogenes and Aspergillus brasiliensis and vitrified embryos stored in Cryotop (closed and naked) and ministraw devices were assessed. In experiment 2, cross-contamination between artificially infected fresh embryos, vitrified and stored in Cryotop (closed and naked) to liquid nitrogen was assessed.”

Line 120: can you be more clear on how the base medium was done?

It has been clarified. “(BM; Dulbecco's phosphate-buffered saline containing calcium chloride [9.0 mM] and magnesium chloride [4.9 mM] and supplemented with 0.2% [wt/vol] BSA at pH 7.2)”

Line 113: please add a reference here on the IETS criteria

A reference has been included ([23] International Embryo Transfer Society. Manual of the International Embryo Transfer Society. In IETS Manual; Stringfellow, D.A., Seidel, S.M., Eds.; Savoy: Champaign, IL, USA, 1998.)

Line 124: how many embryos did you allocate per cryotop or mini-straw? Indicate please

A total of 6 embryos were stored by the device. It is included in Line 126 for experiment 1 and Line 151 for experiment 2.

Line 172: 100 L of suspension = 100 Liters???

It was a mistake. A total of 100 µL of suspension from each sample was inoculated

Figure 2J: it would be nice if you can show the hatching/fully hatched embryos instead since you got a good rate of embryo hatching

In rabbits, intraoviductal mucin coat is crucial allowing, embryo implantation, as it takes place after the remodelling of the embryonic coatings during blastocyst expansion in the uterine horns. Mucin coat deposition is limited to the oviduct for 3 days following ovulation, and the molecular mechanisms of coat material deposition are largely unknown (Denker and Gerdes, 1979). For these reasons, rabbit blastocyst hatched from pellucid zone, but they get trapped inside the mucin layer.

Denker, H.W., Gerdes, H.J. The dynamic structure of rabbit blastocyst coverings. I: transformation during regular preimplantation development. Anatomy and Embryology. 157, 15-34 (1979). 

In both experiment 1 and 2, the number of technical replicates should be reported. How many times did you replicate the experiment? Did you do it in different days?

The experiment was developed for more than a year, so the study had no replication experiment. To store all the embryos, and to reduced bias effect, the embryos were stored in 2 sessions (2 consecutive days) for each experiment. After one year, embryos were thawed on the same day for each experiment

In the discuss it could be important to compare and report the results of embryo hatching of the thawed embryos with data on hatching of fresh embryos from literature. This is something that would add value to the manuscript

A comparison between vitrified and fresh in vitro embryo development has been included.

Reviewer 3 Report

Authors run an interesting study about the contamination risks of embryos when stored under naked and close devices as well as from contaminated LN2, and cross-contamination of sterile LN2 from infected embryos. Besides, authors study the effects of antibiotics-free media on culture development of infected embryos concluding that these media do not support the development of infected embryos whereas media supplemented with antibiotic do support embryo development of infected rabbit embryo.

The long-time effect of contamination is a very interesting parameter for scientist working on the field.

It is interesting how infected embryos reached a so high blastocyst rate (over 70% in many cases). Did the authors transfer these embryos to does?

Electron microscopy showed that infection is caught on embryo surface. Do authors have any data about likely embryo implantation once hatching is produced?

Some points to be expanded/clarified:

Ln 89: what kind of testing was performed to confirm absence of pathogens? Please explain and clarify.

Ln 102-104: were the microorganism added to the tank already full of LN2 or was the tank empty when microorganism were added? Please, clarify the order of infection and LN2 filling of the tank.

Ln 112: what was the PBS temperature for flushing?

Ln 126-129: was sterile LN2 provided by manufacturer tested for pathogens before being used for the experiments? Where the artificially infected tanks refilled with filter LN2 or infected LN2?

Ln 170: do you mean a biosecurity cabinet type II, right?

Ln 172: 100 microliters or liters?

Ln 218-219: what do you mean by the sentence “It is significant that these microorganisms survive in LN2 without cryopreservation”?

Figure 2

Review legend. Picture E) is not mentioned and C, D figures description is incorrect.

Add scale bar in figures F-J.

Clarify that image J corresponds to non-contaminated vitrified-warmed embryos (control well).

Figure 3

Scale bar is hardly seen. Pleas improve view.

Minor comments

The manuscript if plenty of English mistakes and grammar errors that need to be corrected. Submit the manuscript to a native professional for review. For example:

Ln 83: evaluate, not evaluated

Ln 91: ... were tested to be discarded as a source…

Ln 104: …was spilled…

Ln 147-148: …which enables to obtain LN2 from the bottom…

Ln 148: …LN2 was withdrawn…

Ln 149: …and evaporated at room….

Ln 240: positive (no plural)

Author Response

It is interesting how infected embryos reached a so high blastocyst rate (over 70% in many cases). Did the authors transfer these embryos to does?

We thoroughly understand the questions asked by the reviewer. We have done similar items after seeing our data. Nevertheless, initially, no in vivo experiment was proposed due to the uncertainty that this could generate. We are aware that understanding if these pathogens could infect a female during pregnancy, would be necessary.

Electron microscopy showed that infection is caught on embryo surface. Do authors have any data about likely embryo implantation once hatching is produced?

 As previously

Some points to be expanded/clarified:

Ln 89: what kind of testing was performed to confirm absence of pathogens? Please explain and clarify.

After disinfection, we tested biobanks by culture method. It has been indicated in the manuscript.

Ln 102-104: were the microorganism added to the tank already full of LN2 or was the tank empty when microorganism were added? Please, clarify the order of infection and LN2 filling of the tank.

Growth and propagation of bacteria and fungus strains were performed according to manufacturer recommendation at an infective titer of 106 CFU/mL for each microorganism. Then, 1 ml of each microorganism was atomized directly on the liquid nitrogen in the half-full biobank. After that, biobank was filled and stored one week until the embryos were incorporated.

Ln 112: what was the PBS temperature for flushing?

The temperature was included in the manuscript (Line 124:  25ºC)

Ln 126-129: was sterile LN2 provided by manufacturer tested for pathogens before being used for the experiments? Where the artificially infected tanks refilled with filter LN2 or infected LN2?

Effectively. Before embryo vitrification procedure, both biobanks and the nitrogen from the supplying company were evaluated. Besides, 11 liquid nitrogen samples from the supplier company were assessed across the year (once a month). In none of the analyzes did we find any organism object of the study (viable cultivable). It was included in the manuscript in Line 168-169.

Ln 170: do you mean a biosecurity cabinet type II, right?

Sure thing!

 Ln 172: 100 microliters or liters?

It was a mistake. A total of 100 µL of suspension from each sample was inoculated

Ln 218-219: what do you mean by the sentence “It is significant that these microorganisms survive in LN2 without cryopreservation”?

This phrase was removed. It sound redundant.

Figure 2

Review legend. Picture E) is not mentioned and C, D figures description is incorrect. Add scale bar in figures F-J. Clarify that image J corresponds to non-contaminated vitrified-warmed embryos (control well).

 All suggested changes have been implemented

Figure 3

Scale bar is hardly seen. Pleas improve view.

We have modified the size of the figure for a better vision. Besides, magnification is included in the legend.

Minor comments

The manuscript if plenty of English mistakes and grammar errors that need to be corrected. Submit the manuscript to a native professional for review.

An English professional service has reviewed the manuscript (NM Language Services)